# Enhanced Skills Training in Affective and Interpersonal Regulation (ESTAIR): A New Modular Treatment for ICD-11 Complex Posttraumatic Stress Disorder (CPTSD)

**DOI:** 10.3390/brainsci13091300

**Published:** 2023-09-09

**Authors:** Thanos Karatzias, Edel Mc Glanaghy, Marylene Cloitre

**Affiliations:** 1School of Health & Social Care, Edinburgh Napier University, Edinburgh EH11 4BN, UK; 2NHS Lothian Rivers Centre, EH11 1BG, Stanford University, 450 Jane Stanford Way, Stanford, CA 94305, USA; 3NHS Forth Valley, Mayfield Building, Falkirk Community Hospital, Scotland FK1 5QE, UK; edel.mcglanaghy@nhs.scot; 4National Centre for PTSD, Dissemination and Training Division, VA Palo Alto Health Care System, 795 Willow Road, Menlo Park, CA 94025, USA; marylenecloitre@gmail.com; 5Department of Psychiatry and Behavioral Sciences, Stanford University, Palo Alto, CA 94305, USA

**Keywords:** complex PTSD, CPTSD, modular treatment, ESTAIR

## Abstract

ICD-11 Complex Posttraumatic Stress Disorder (CPTSD) is a relatively new condition; therefore, there is limited available evidence for its treatment. Prior to the recognition of CPTSD as a separate trauma condition, people who met criteria were often diagnosed with multiple co-morbid conditions such as PTSD, anxiety, depression, and emotional dysregulation difficulties. In the absence of a coherent evidence base, treatment tended to involve multiple treatments for these multiple conditions or lengthy phase-based interventions, often delivered in an integrative fashion, which was not standardized. In this paper, we present Enhanced Skills Training in Affective and Interpersonal Regulation (ESTAIR), a new flexible multi-modular approach for the treatment of CPTSD and its transdiagnostic symptoms. ESTAIR is consistent with trauma-informed and patient-centered care, which highlights the importance of patient choice in identification and sequencing in targeting CPTSD symptoms. Directions for future research are discussed.

## 1. ICD-11 PTSD and CPTSD

CPTSD has been formally introduced into the diagnostic nomenclature in the eleventh edition of the *International Classification of Diseases and Related Health Problems* [1]. Given that CPTSD is a new disorder, there is little evidence of effective treatments. There is substantial evidence on the effectiveness of interventions for DSM-IV PTSD, but there is little evidence whether these interventions are equally effective for ICD-11 CPTSD, or which interventions may be optimal for the treatment of this debilitating condition. The purpose of this paper is to present the rationale for offering a flexible, multi-modular patient-centred approach to the treatment of CPTSD, namely Enhanced Skills Training in Affective and Interpersonal Regulation (ESTAIR), as well as its feasibility and potential benefits. The paper concludes with recommendations for further research on the effectiveness of ESTAIR for CPTSD.

## 2. Definition of PTSD and CPTSD

Exposure to a traumatic event is a prerequisite for consideration of either ICD-11 PTSD or CPTSD. The diagnostic criteria for PTSD consist of three symptom clusters that relate specifically to the traumatic event, including re-experiencing in the here and now, avoidance of traumatic reminders, and heightened sense of threat. Functional impairment must also be present to meet diagnostic criteria. The diagnosis of CPTSD comprises six symptom clusters, the three PTSD clusters listed above and three symptom clusters representing pervasive and chronic disturbances in self-organization (DSO), including affect dysregulation, negative self-concept, and difficulties in forming and maintaining relationships. Functional impairment resulting from the PTSD symptom cluster and the DSO are also necessary to meet diagnostic criteria for CPTSD [1].

ICD-11 introduced CPTSD as a diagnosis distinct from PTSD acknowledging the effect that chronic, repeated, or severe interpersonal trauma can have on self-organization-related mechanisms. Exposure to traumatic events, which are prolonged and from which escape is difficult or impossible, is more likely to lead to CPTSD than PTSD. Such traumatic events include experiences such as repeated childhood sexual or physical abuse, domestic violence, prolonged combat exposure, torture, and genocide campaigns [1].

Distinct from earlier conceptualizations of complex PTSD [2], the presence of prolonged trauma or early life traumas is a risk factor, not a prerequisite for CPTSD and any traumatic stressor can lead to either PTSD or CPTSD. This recognizes the role of personal (e.g., genetic or dispositional) and environmental (e.g., social stressors and support) risk and protective factors for vulnerability to the disorder. Diagnosis is therefore based on the presence of CPTSD symptoms in the context of the individual’s personal history.

## 3. Evidence Supporting the Distinction between CPTSD versus PTSD

The selection of symptoms in the PTSD cluster (i.e., re-experiencing, avoidance, and sense of threat) is supported by decades of research [3]. The identification of the DSO symptoms of affect dysregulation (e.g., having problems calming down, feeling emotionally numb), negative self-concept (e.g., feeling worthless or like a failure), and relationship disturbances (e.g., having difficulty feeling close to others or maintaining relationship) was based on results from DSM-IV field trials, which investigated the most frequently reported CPTSD symptoms [4] and results from an expert opinion survey where clinicians were asked to identify the most common and impairing CPTSD symptoms [5].

This two-factor formulation of CPTSD has been confirmed in several studies all over the world and across general population and clinical samples [6] and it appears to be easy for clinicians to correctly identify as indicated by high rates of accurate differential diagnosis between CPTSD and PTSD and as compared to normality among over 1700 clinicians in 73 countries [7].

Traumatic stressors, especially in childhood, can increase vulnerability for several conditions later in life [8]. Thus, it is not surprising that CPTSD symptoms can be present in a number of other conditions (e.g., psychosis; [9,10]) and indeed, prior to the introduction of CPTSD people often attracted a wide range of diagnoses to account for their wide-ranging symptoms. Therefore, it is of significant interest to explore whether targeting and treating CPTSD symptoms in other conditions can result in symptom reduction overall and perhaps increase our understanding of the mechanisms of distress in other co-morbid conditions associated with a trauma history.

## 4. Treating CPTSD Using Effective Treatments for PTSD

The ICD-11 separation of post-traumatic reactions into two rather than one disorder is consistent with a personalized medicine approach to care and treatment. ICD-11 CPTSD has a greater number of symptoms typically resulting from multiple, interpersonal, chronic, and/or childhood problems [11]. Several studies comparing ICD-11 PTSD to ICD-11 CPTSD have indicated that CPTSD is associated with significantly poorer functioning, greater comorbidity and poorer quality of life compared to ICD-11 PTSD (e.g., [12,13]). CPTSD is a more common condition than PTSD in trauma-exposed population-based studies (e.g., 5.3% vs. 12.9% in the UK, [12]), general population-based samples (e.g., 3.4% vs. 3.8 in the USA, [13]) and clinical trauma samples (e.g., 75.6% vs. 24.4 in the UK, [11]). Considering that CPTSD comprises a greater number of different types of symptoms and is more severe than PTSD in clinically meaningful ways (i.e., number of symptoms and impaired functioning), it may be the case that optimal treatment of CPTSD may require a greater number of different kinds of interventions or a longer course of treatment compared to PTSD. Although the development of different treatments may result in better outcomes for PTSD and CPTSD, it is also possible that the same treatments can be used for these two conditions with equally good outcomes.

Existing treatment guidelines for PTSD recommend trauma-focused psychological therapies, such as cognitive-behavioral therapy (CBT), and Eye Movement Desensitization and Reprocessing (EMDR) for PTSD (e.g., National Institute for Clinical Excellence [NICE], [14]; Australian Centre for Posttraumatic Mental Health [ACPMH], [15]). Trauma-focused treatments typically include repeated in vivo and/or imaginal exposure to the trauma and/or reappraisal of the meaning of the trauma and its consequences. However, two recent meta-analyses have suggested that these interventions might be less effective for patients with CPTSD.

In a meta-analysis by Karatzias and colleagues [16], randomized controlled studies of PTSD, which included measures of the three symptom clusters of PTSD, as well as symptoms specific to CPTSD, namely affect dysregulation, negative self-concept, and disturbances in relationships, were reviewed using proxy measures. Analyses assessing outcomes for each of the six specific symptom clusters revealed that, compared to waitlist or treatment as usual, cognitive-behavioral therapies, exposure therapy and EMDR yielded superior outcomes. However, this effect was negatively moderated by a history of childhood abuse, where outcomes for each of the six CPTSD symptom clusters were consistently less positive for the subgroup of participants with childhood trauma. A second meta-analysis by Coventry and colleagues [17] also found that both trauma and non-trauma focused therapies provided substantial benefits for PTSD symptoms, while the positive effects for the DSO symptom clusters were modest. It was also suggested that multi-component therapies led to better outcomes, particularly among survivors of childhood sexual abuse. This meta-analysis also reported that a likely CPTSD diagnosis was associated with higher attrition rates compared with PTSD. Overall, these two meta-analyses suggest that existing therapies routinely used for PTSD may be less optimal for those likely to have CPTSD and that multicomponent modular therapies can be useful for those with CPTSD.

## 5. Patient-Centred Care and CPTSD

Patient-centred care advocates that symptoms or difficulties that are important to one individual might not necessarily be important for another individual with the same condition. The principles of patient-centred care also highlight that the needs of people with mental health problems may change over time while they receive treatment and care from services [18]. The Health Foundation [18] has identified a framework that comprises four principles of person-centred care, including offering dignity, compassion and respect, coordinated care, personalised care and empowerment, resilience, and agency.

In this framework, the role of the health care professional is to enable the individual to make decisions about their own care and treatment based on their needs at the time they access help and support. This approach is also aligned with the trauma-informed principles of empowerment and collaboration [19]. Several health and economic outcomes have been associated with person-centred care including less use of emergency services [20], greater treatment adherence [21], greater patient satisfaction [22], and increased staff performance and morale [21].

The use of flexible modular and sequential treatments, where therapeutic targets and interventions are offered in line with patient recovery stage, needs, and preferences are consistent with the principles of person-centred care. Using person-centred care might also result in reduced attrition rates, which can be a particular issue in those with CPTSD [17], although this is still an empirical question. In conclusion, person-centred treatment approaches implemented through the use of flexible modular therapies hold the promise of being an efficient and effective strategy for optimizing outcomes for those with CPTSD [23].

## 6. The Development of STAIR Narrative Therapy for PTSD and Other Complex Forms of Traumatization

Skills Training in Affective and Interpersonal Regulation (STAIR) followed by narrative therapy (NT) is a two-module treatment that was developed, prior to the introduction of CPTSD to address the multiple problems that individuals with histories of childhood abuse may develop. This was based on a substantial literature base indicating that in addition to PTSD, these individuals presented with significant problems in emotion regulation and relational capacities (see [24]).

The rationale for the treatment was to directly target and address these difficulties as well as the PTSD symptoms. STAIR included interventions that helped individuals become more skilled in experiencing, managing and modulating the expression of feelings (e.g., using feelings wheel, written descriptions of feelings, focused breathing) and to become more flexible in interpersonal expectations (e.g., cognitive re-appraisal), more skilled in communication (e.g., effective assertiveness) and more comfortable and trusting in relationships (e.g., practicing positive activities and engaging in social support). The Narrative Therapy module was an adapted version of exposure therapy (i.e., prolonged exposure), in which the participant recounted memories of their traumatic experiences with the goal of reducing fear reactions to these events and re-appraising the meaning of the experience(s) in a more adaptive manner.

The treatment was sequenced so that STAIR preceded NT. The primary goal of the STAIR modules was to improve overall functioning in day-to-day life. A secondary goal was to use the time during STAIR to establish a stable therapeutic alliance and emotion regulation skills both of which were expected and indeed demonstrated to facilitate the narrative work [25]. Three randomized controlled trials [26,27,28] indicated that STAIR Narrative Therapy provided significant improvements and large effect sizes in regard to PTSD, emotion regulation, and interpersonal difficulties.

Over time, and particularly with the introduction of the CPTSD diagnosis, it became evident that the symptom profile, which was first observed among individuals with early life trauma was also applicable to individuals with adult-onset traumas of a sustained nature including child soldiers [29], prisoners of war [30], refugees [31] and individuals experiencing sustained community violence [32]. The STAIR-NT protocol has therefore been adapted and revised to address the full range of symptom clusters comprising the diagnosis of CPTSD, to be appropriate to the wide range of trauma-exposed populations experiencing CPTSD, and to attend to the principles of personalized care.

The revised protocol, now called Enhanced STAIR (ESTAIR), comprises four modules: Emotion Regulation, Relationship Patterns, Self-Concept, and Narrative Therapy. The treatment is proposed to be applicable to a wide range of trauma-exposed populations with CPTSD such as veterans, refugees, individuals who have experienced domestic violence, or those with childhood trauma. Lastly, and most importantly, the four modules are expected to be used in a flexible order and a flexible duration to be responsive to the needs and preferences of the patients.

## 7. ESTAIR: A New Modular Person-Centred Therapy for CPTSD

ESTAIR has adopted all the theoretical and clinical principles of STAIR Narrative Therapy but in line with the ICD-11 formulation of CPTSD, ESTAIR includes modules to treat all symptom clusters of CPTSD, as detailed in Table 1 below. The essential principle of STAIR Narrative Therapy is that trauma recovery involves not only attention to memories of traumatic events from the past, but also covers the impact of trauma on the present as it impacts current relationships, emotional distress in day-to-day life and quality of life. Accordingly, the program includes traditional interventions related to processing of the trauma memories (e.g., reappraisal of their meaning) as well as practical skills training and related interventions to improve relationships, sense of self, emotion regulation, and mood management. 

ESTAIR is a flexible modular approach for the treatment of CPTSD, where patient and therapist collaborate on the selection of a set treatment modules intended to resolve specific problems of concern, based on the stage of recovery the person is at, their preference and the symptoms associated with most distress at that particular moment when they access treatment and support.

Each of the four ESTAIR modules is structurally equivalent (6 sessions each). The Affect Dysregulation module focuses on skills training in relation to identifying and labelling feelings, emotion management, distress tolerance, and acceptance of feelings and experiencing positive emotions. The Negative Self-concept module focuses on the impact of trauma on one’s self concept, how to stay in the present moment and combat dissociation, cultivate self—compassion and mindfulness skills, challenge thinking patterns including tackling negative thoughts rules and assumptions that relate to ones-self, how to be more nurturing towards oneself, explore personal qualities, and develop a balanced view of self. The Disturbed Relationships module focuses on exploration and revision of maladaptive interpersonal schemas, effective assertiveness, awareness of social context, and flexibility in interpersonal expectations and behaviours that are displayed in social interactions. The Narrative Therapy module begins with an organization of a memory hierarchy in which the individual identifies several key traumatic memories. The remaining sessions are dedicated to telling the story of selected experiences and a re-appraisal of the meaning of the event. Interventions associated with the trauma-processing work include flexible use of adjunctive interventions such as systematic journaling about the experience and visiting and re-evaluating the experience at the site of the trauma. Importantly, both therapist and patient contrast and compare old trauma-generated beliefs to the newly emerging and more positive sense of self and perspectives on others [33].

## 8. Limitations of ESTAIR as an Intervention for CPTSD

Although ESTAIR has a number of advantages for the treatment of CPTSD, it also presents with several limitations, which are common across all flexible multi-modular approaches. Firstly, there may be greater reliance on clinician’s skill in building a therapeutic alliance with the patient in order to enable decisions on the sequencing of modules. Although ESTAIR modules can be offered in a set order, this would be against the principles of person-centred care. It would be essential to have a discussion very early in the treatment about identifying appropriate treatment goals and symptom targets on the basis of the recovery stage that a person is in, risk of decompensation and current symptom prominence. Emerging evidence suggests that slightly different CPTSD symptom profiles might emerge from difference traumatic stressors in terms of symptom prominence [34] and this information can also guide the sequencing of modules. Secondly, the application of ESTAIR also requires routine assessment of symptoms for feedback and decision making, which may be viewed as burdensome by some services and clinicians, but which may actually reduce the likelihood of dropout [35]. Considering that the application of ESTAIR requires therapists making judgments about the sequencing of different modules, it is essential to provide appropriate support at the service level, including the use of clinical supervision to enable clinician’s decision making.

## 9. Directions for Future Research Using ESTAIR for the Treatment of CPTSD

This paper has presented ESTAIR, a promising person-centred flexible treatment approach and provided the rationale for its usefulness for people with CPTSD. However, there is clearly a need for further research in the field. Firstly, it is essential to assess whether ESTAIR is a suitable intervention not only for CPTSD but also PTSD. Although this is still an empirical question, it is expected that fewer sessions or modules may be required for PTSD. There is also a need to compare ESTAIR with existing interventions routinely used for PTSD to explore optimal approaches for the treatment of CPTSD. Some recent evidence suggests that combination of modular therapies, i.e., STAIR plus established PTSD treatments such as prolonged exposure are not superior to prolonged exposure alone for the treatment of CPTSD related to childhood abuse [36]. Although this evidence might raise some concerns about the usefulness of modular therapies for CPTSD, it might also be useful to conduct further research in these areas using ESTAIR as opposed to STAIR across different populations with CPTSD. Furthermore, comparing the effectiveness of ESTAIR vs. existing therapies routinely used for PTSD, such as prolonged exposure or EMDR in those with CPTSD should include t numerous outcomes addressing not only symptom reduction, but also drop-out rates and patient satisfaction. Future research should also explore the effectiveness of adaptation of treatments for other disorders or from other traditions such as Interpersonal Psychotherapy (IPT) and mindfulness therapy or the comparative effectiveness of ESTAIR against these interventions.

Furthermore, it would also be useful to explore different kinds of sequencing strategies for the ESTAIR modules of where, for example, in one condition patient choice drives the sequencing compared with a fixed sequence. At present, there is mixed evidence for the impact of patient choice on therapy effectiveness (e.g., for depression [37]). Another design, namely “sequential, multiple assignment randomized trials” or SMART [38], would involve open-ended treatment sequences, where the selection of the next module is determined by the patient’s response to previous module. Furthermore, there is a need to assess the effectiveness and acceptability of ESTAIR in real life clinics using pragmatic designs with fewer inclusion and exclusion criteria and across different populations with CPTSD including community populations, prisoners, refugees, and veterans. Considering that the DSO symptoms of CPTSD are cross diagnostic and present in a number of conditions that might also result from traumatic stressors, it would also be essential to explore whether ESTAIR is a beneficial treatment for comorbid CPTSD symptoms that can be present in other conditions such as psychosis, and whether actively targeting these symptoms can have an impact on symptom severity of the primary diagnosis. Finally, considering that there may be some cultural variations in the DSO symptoms of CPTSD [39], ESTAIR may require cultural adaptations in its content to enhance its cross-cultural relevance. We now have an opportunity to target specific and effective therapies to those who need them most and the CPTSD diagnosis allows us to be more precise about how we can best do this, in a trauma-informed, person-centred manner.

## Figures and Tables

**Table 1 brainsci-13-01300-t001:** ESTAIR modules and session content.

Formulation Session	
Understanding difficulties and preparing for change. *During session 1, one of the four modules is collaboratively chosen to begin with.*	
Emotion Regulation	Sense of Self
1. Introduction and Emotional Awareness 2. Focus on the Body 3. Focus on Thoughts 4. Focus on Behaviours 5. Distress Tolerance 6. Accomplishments and Summary of Work	1. What is the ‘Self’ 2. Me in the Moment 3. Thinking about Self 4. Criticism and Compassion 5. Who am I, In Relation to Others 6. Accomplishments and Summary of Work
Relationship Patterns	Narrative Reprocessing
1. Understanding Relationship Patterns 2. Increasing Assertiveness 3. Managing Power with Respect 4. Increasing Respect for Yourself and Others 5. Increasing Closeness 6. Summary of Skills and Accomplishment	1. Introduction to Narrative Reprocessing 2. Narrative of the Most Distressing Memory 3. Continuation of Narrative Reprocessing 4. Continuation of Narrative Reprocessing 5. Continuation of Narrative Reprocessing 6. Relapse Prevention and Summary of Work

## Data Availability

Not applicable.

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
