# Peer review of "Enhanced Skills Training in Affective and Interpersonal Regulation (ESTAIR): A New Modular Treatment for ICD-11 Complex Posttraumatic Stress Disorder (CPTSD)"

_brainsci, 2023, doi:10.3390/brainsci13091300_

Round 1

Reviewer 1 Report

The topic of this work is undeniably intriguing. Although PTSD is widely discussed, CPTSD seems to be overlooked. It is imperative for the public to have a comprehensive understanding of the effects and treatment of CPTSD. Nevertheless, the content of this study needs improvement by providing more related information.

Incorporating information on the prevalence of CPTSD and PTSD in the background can be beneficial. This data can encompass global statistics or specific countries and regions.

Regarding the section on the Definition of PTSD and CPTSD, more previous studies/ information can be added (cited) to support the content.

In the section of ESTAIR: A new modular person-centred therapy for CPTSD, it was stated “ESTAIR has adopted all the theoretical and clinical principles of STAIR Narrative Therapy…” What are the theoretical principles and clinical principles of STAIR Narrative Therapy used? Some description of these principles can be provided.

It is crucial to emphasize the significance of this study and thoroughly examine its potential contributions to various perspectives.

Minor issues:
Please ensure that the complete form of an acronym is provided when it is first introduced in the text. This will allow readers to understand the meaning of the acronym and avoid confusion. For instance, in the initial sentence of the opening paragraph, please use "Complex Post Traumatic Stress Disorder (CPTSD)" instead of just "PTSD". This will enhance clarity and comprehension for the readers.

The conclusion and direction for future research can be separated into two sections.

Author Response

We would like to thank the reviewer for the comments and we have made appropriate amendments as follows:

The topic of this work is undeniably intriguing. Although PTSD is widely discussed, CPTSD seems to be overlooked. It is imperative for the public to have a comprehensive understanding of the effects and treatment of CPTSD. Nevertheless, the content of this study needs improvement by providing more related information.

Thank you for the positive comments.

Incorporating information on the prevalence of CPTSD and PTSD in the background can be beneficial. This data can encompass global statistics or specific countries and regions.

We have additional information on the prevalence of PTSD and CPTSD as requested by the reviewer.

Regarding the section on the Definition of PTSD and CPTSD, more previous studies/ information can be added (cited) to support the content.

In this section we have attempted to cite relevant reviews e.g. 

Brewin, C.R., Cloitre, M., Hyland, P., Shevlin, M., Maercker, A., Bryant, R.A., Humayun, A., Jones, L.M., Kagee, A., Rousseau, C. and Somasundaram, D., 2017. A review of current evidence regarding the ICD-11 proposals for diagnosing PTSD and complex PTSD. Clinical psychology review, 58, pp.1-15.

which summarize important literature in this area.

In the section of ESTAIR: A new modular person-centred therapy for CPTSD, it was stated “ESTAIR has adopted all the theoretical and clinical principles of STAIR Narrative Therapy…” What are the theoretical principles and clinical principles of STAIR Narrative Therapy used? Some description of these principles can be provided.

We have added some additional information on the principles of ESTAIR as recommended by the reviewer. 

It is crucial to emphasize the significance of this study and thoroughly examine its potential contributions to various perspectives.

This is being addressed in the introduction.

Minor issues:
Please ensure that the complete form of an acronym is provided when it is first introduced in the text. This will allow readers to understand the meaning of the acronym and avoid confusion. For instance, in the initial sentence of the opening paragraph, please use "Complex Post Traumatic Stress Disorder (CPTSD)" instead of just "PTSD". This will enhance clarity and comprehension for the readers.

This has been rectified in the revised version of the manuscript.

The conclusion and direction for future research can be separated into two sections.

We have revised the title of this section to prevent confusion.

Reviewer 2 Report

The manuscript describes ESTAIR as a new modular treatment for CPTSD, this is a new and important topic for discussion, below are mu comments to the authors

·        The authors describe and distinguish between PTSD and CPTSD, however little is known about the types of subjects or the etiology behind developing experiencing CPTSD, for example, it is well-established war-displaced refugees experience PTSD, do they suffere from CPTSD? Is there any evidence about that? Also, are here any validated scales to screen and distinguish PTSD from CPTSD that can be applied in epidemiological studies?

·        The authors describe treatments used for PTSD that can be effective for CPTSD such as CBT and others. Although the manuscript is focused on the non-pharmacological intervention, however, to ensure a comprehensive presentation, I think that the author need to highlight any potential role for the pharmacological intervention in CPTSD

·        The authors describe STAIR and ESTAIR for PTSD and CPTSD respectively, I suggest that the author add tables to summarize their effectiveness by presenting the mentioned trials, their design, sample size, duration, etc..

·        Some typing errors such as “Table 1 about Here” and others should be corrected

·         

Author Response

We would like to thank the reviewer for their comments and we have made appropriate amendments as follows:

The manuscript describes ESTAIR as a new modular treatment for CPTSD, this is a new and important topic for discussion, below are mu comments to the authors

We would like to thank the reviewer for the positive comments.

The authors describe and distinguish between PTSD and CPTSD, however little is known about the types of subjects or the etiology behind developing experiencing CPTSD, for example, it is well-established war-displaced refugees experience PTSD, do they suffere from CPTSD? Is there any evidence about that? Also, are here any validated scales to screen and distinguish PTSD from CPTSD that can be applied in epidemiological studies?

Information regarding the nature of PTSD and CPTSD is provided in the section "Definition of PTSD & CPTSD". The international trauma questionnaire (ITQ) and the International Trauma Interview (ITI) have been developed for the assessment of PTSD and CPTSD although this information is beyond the scope of the present paper. 

The authors describe treatments used for PTSD that can be effective for CPTSD such as CBT and others. Although the manuscript is focused on the non-pharmacological intervention, however, to ensure a comprehensive presentation, I think that the author need to highlight any potential role for the pharmacological intervention in CPTSD

We focused on psychological interventions in this manuscript as clinical guidelines (see p. 3, 2nd paragraph) recommend psychological treatments for trauma as a first line of support. The usefulness of pharmacotherapies for PTSD is beyond the scope of this paper.

The authors describe STAIR and ESTAIR for PTSD and CPTSD respectively, I suggest that the author add tables to summarize their effectiveness by presenting the mentioned trials, their design, sample size, duration, etc..

For the sake of brevity, we have decided not to report this detail. However, the available references have been provided for readers if they wish to have more information on the effectiveness of STAIR. 

Some typing errors such as “Table 1 about Here” and others should be corrected

We have checked the manuscript for typing errors as suggested by the reviewer.